# LSH-SMILE: Locality Sensitive Hashing Accelerated Simulation and Learning

**Chonghao Sima**
Department of Computer Science
Purdue University
West Lafayette, IN, USA, 47907
`simac@purdue.edu`

**Yexiang Xue**
Department of Computer Science
Purdue University
West Lafayette, IN, USA, 47907
`yexiang@purdue.edu`

## Abstract

The advancement of deep neural networks over the last decade has enabled progress in scientific knowledge discovery in the form of learning Partial Differential Equations (PDEs) directly from experiment data. Nevertheless, forward simulation and backward learning of large-scale dynamic systems requires handling billions mutually interacting elements, the scale of which overwhelms current computing architectures. We propose Locality Sensitive Hashing Accelerated Simulation and Learning (LSH-SMILE), a unified framework to scale up both forward simulation and backward learning of physics systems. LSH-SMILE takes advantages of (i) the locality of PDE updates, (ii) similar temporal dynamics shared by multiple elements. LSH-SMILE hashes elements with similar dynamics into a single hash bucket and handles their updates at once. This allows LSH-SMILE to scale with respect to the number of non-empty hash buckets, a drastic improvement over conventional approaches. Theoretically, we prove a novel bound on the errors introduced by LSH-SMILE. Experimentally, we demonstrate that LSH-SMILE simulates physics systems at comparable quality with exact approaches, but with way less time and space complexity. Such savings also translate to better learning performance due to LSH-SMILE's ability to propagate gradients over a long duration.

## 1 Introduction

Learning-driven scientific discovery has enjoyed rapid progress thanks to the advancement of deep neural networks over the last decade. Since Partial Differential Equations (PDEs) are widely used to model physics systems, a fruitful line of research has been developed focusing on learning PDEs from experimental data, including Finzi et al. [2020], Greydanus et al. [2019], Matsubara et al. [2020].

Nevertheless, successful applications of machine learning for scientific discovery still face multiple challenges, many of which are computational. Both the forward simulation and backward learning of large-scale dynamic systems requires handling billions mutually interacting elements, the scale of which overwhelms current computing architectures. Forward simulation involves simulating the trajectory of a PDE system from a given starting state. Recent work of Greydanus et al. [2019] demonstrates such process can be represented using a neural network similar to ResNet. Backward learning is to discover (the parameters of) the physics model automatically from experimental data, which also attracts recent attention in Niu et al. [2020] and Xue et al. [2021]. Backward learning can be achieved by embedding a neural network modeling the forward simulation into the overall architecture and minimizing a loss function which penalizes the difference between the simulated result and the observed data via back-propagation. In both forward simulation and backward learning, billions of mutually interacting elements resulted from applying the finite difference or finite element

approaches have to be handled efficiently. Current brute-force approaches which treat each element as one matrix element do not scale to meet the computational need.

To tackle the computational bottleneck, we propose **L**ocality **S**ensitive **H**ashing Accelerated **Sim**ulation and **Le**arning (LSH-SMILE), a unified framework to scale up both forward simulation and backward learning of physics systems represented in a set of PDEs. The method is based on three key observations. The first is that only a small fraction of elements change while the majority of elements remain the same in one step forward simulation. The second is that the one-step update of one element depends on the values of a small set of neighboring elements. The third is that many elements share similar temporal dynamics. The three observations are shared among the simulations of a wide variety of physics phenomena, especially in the so-called *interface problems* that have applications in fluid dynamics, heat transfer, cracking formation, etc. Motivated by these observations, LSH-SMILE harnesses Locality Sensitive Hashing (LSH) to boost the computational throughput. By representing each element as a vector of its value and its neighbor elements' value, we hash elements into hash buckets via LSH, where elements with similar self and neighboring element values are grouped into the same bucket. Then the forward simulation and backward learning are carried out for all the elements hashed into the same bucket at once. In this way, the time and space complexities of LSH-SMILE are reduced to be propotional to the number of non-empty hash buckets, a drastic improvement compared to the number of distinct elements in the brute-force algorithm. We also prove a novel bound on the quality of the approximation of LSH-SMILE and provide an interface to control the error rate. LSH-SMILE explores a novel way of using LSH in learning and simulation. Compared to traditional LSH usage case in the nearest neighbor search where the hash table stays the same, LSH-SMILE needs to maintain a hash table that changes dynamically over time. New theoretical and experimental ideas can be sparkled via exploring this new usage of LSH.

Experimentally, LSH-SMILE is able to simulate and learn physics systems represented in PDEs at a comparable quality with the exact methods while saving drastically in computation. We focus our attention on systems in nano-physics, where we simulate and learn the grain growth in materials (Fan and Chen [1997]) and model the spatial temporal dynamics of void shaped defects, namely nanovoids, in materials under high temperature and irradiation (Millett et al. [2011]). The proposed LSH-SMILE reduces the overall simulation time compared to an exact Torch implementation by 70% in the grain growth simulation, and by 95% in the nanovoid simulation. In terms of precision, LSH-SMILE matches the ground truth results to the scale of $10^{-5}$ in both the nanovoid and the grain growth simulations after 100 simulation steps. Such savings directly translate to better learning performance. In our experiment which uses LSH-SMILE to learn the physics parameters governing a grain growth process, LSH-SMILE-based learning approach can successfully identify the correct model parameters while brute-force baseline approaches cannot, The success is due to a longer forward simulation (30 time steps) embedded in the backward learning process enabled by the computational savings of LSH-SMILE. Baseline approaches can only embed 10 forward simulation steps with similar computational budget. One related approach is the Fast Multi-pole Method (Rokhlin [1985]) used in astrophysics. FMM is used to calculate long-ranged forces in the n-body problem, by grouping elements which are close in distance into a single source. However, LSH-SMILE groups elements based on their similarity in future updates. We believe this is a key difference. Overall, the computational innovation of LSH-SMILE opens the door for faster and better scientific discoveries.

## 2 Preliminaries

**Energy-based systems represented in Partial Differential Equations (PDEs)**. The dynamics of many physics systems can be described with first or second-order Partial Differential Equations (PDEs) modeling the system energy. Given the state $\vec{u}(\vec{p}, t) = (u_1(\vec{p}, t), \ldots, u_m(\vec{p}, t))^T$, where $\vec{p} = (x_1, \ldots, x_d)^T$ denotes the spatial coordinates and $t$ denotes time, such PDEs are in the form:

$$\frac{\partial \vec{u}(\vec{p}, t)}{\partial t} = D(\vec{u}) + G(\vec{u})\nabla F(\vec{u}) + I(\vec{u})\nabla^2 H(\vec{u}). \tag{1}$$

Here, $F$ and $H$ are functions that map $\mathcal{R}^m$ to $\mathcal{R}$. $\nabla = (\frac{\partial}{\partial x_1}, \ldots, \frac{\partial}{\partial x_d})^T$ represents the gradient operator, $\nabla^2 = \nabla \cdot \nabla$ is the Laplace operator. $D$ is a function that maps $\mathcal{R}^m$ to $\mathcal{R}^m$. $G(\vec{u})$ is an $m$-by-$d$ matrix and $I(\vec{u})$ is a $m$-by-1 vector.

Many notable physics systems can be represented using PDEs in Equation 1. The widely used Allen-Cahn and Cahn-Hilliard Equations in phase field modeling are good examples. The Cahn–Hilliard

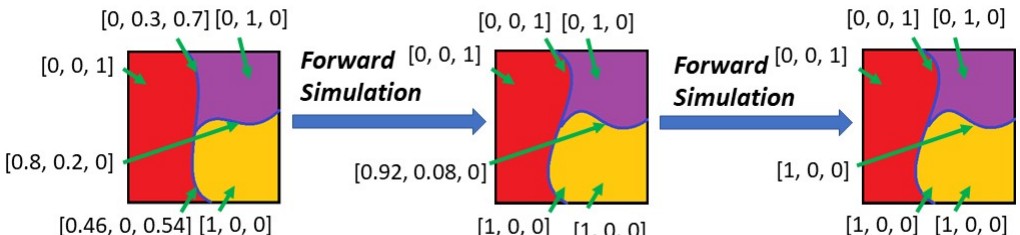

Figure 1: A toy example showing the forward simulation of grain growth. Here, the state $\vec{u}(\vec{p}, t)$ is represented using a triple $(u_1(\vec{p}, t), u_2(\vec{p}, t), u_3(\vec{p}, t))$. $\vec{u}$ vary over location $\vec{p}$ and time $t$ according to a specific form of Equation 4. $u_1$ ($u_2$, $u_3$) is 1 if inside the first (second, third) grain and 0 outside. Values between 0 and 1 for $u_1$, $u_2$, $u_3$ can be found at the boundary.

equation and Allen-Cahn equation are used to describes the physics process of field variables in nano-scale materials. The values of the field variables represent the micro-structure composition at different spatial coordinates. The Cahn–Hilliard equation has the general form:

$$\frac{\partial u}{\partial t} = \nabla \cdot \left( M \nabla \frac{1}{N} \frac{\delta F}{\delta u} \right). \tag{2}$$

The Allen-Cahn equation has the general form:

$$\frac{\partial v}{\partial t} = -L \frac{\delta F}{\delta v}. \tag{3}$$

Here $u$ and $v$ are the field variables of interest and they are assumed to be continuous and changing rapidly across the inter-facial regions. $t$ is time. $M$, $N$, $L$ are physics parameters that are related to the changing process. $F$ is the energy function which is different in different applications. More details are left in the supplementary materials.

**Discretization with finite difference.** Finite difference methods allow us to simulate and learn PDEs using the forward and backward propagation of a neural network. It has attracted recent attentions in recent works, e.g., Wang et al. [2020], Dong and Simos [2017], Forsythe and Wasows [1963] and Shojaei et al. [2019]. More specifically, let $\{x_{i,1}, x_{i,2}, \ldots, x_{i,R}\}$ be a finite discretization of the domain of $x_i$, where each $x_{i,k}$ is called a discretization point. $\{t_1, t_2, \ldots, t_S\}$ is the discretization of the time. Let $\vec{i} = (i_1, i_2, \ldots, i_d)$. We use $\vec{u}(\vec{i}, j)$ as a shorthand for $\vec{u}$ evaluated at position $\vec{i} = (x_{1,i_1}, \ldots, x_{d,i_d})^T$ and time $t_j$, i.e., $\vec{u}(\vec{i}, j) = \vec{u}(x_{1,i_1}, \ldots, x_{d,i_d}, t_j)$. Each discretized position $\vec{i}$ is also called an *element*. The left-hand side of Equation 1 can be discretized using the finite difference: $\frac{\partial \vec{u}(\vec{i}, j)}{\partial t} \approx \frac{\vec{u}(\vec{i}, j+1) - \vec{u}(\vec{i}, j)}{t_{j+1} - t_j}$. Similarly, notice $\nabla F(\vec{u}) = (\frac{\partial F(\vec{u})}{\partial x_1}, \ldots, \frac{\partial F(\vec{u})}{\partial x_d})^T$, where $\frac{\partial F(\vec{u})}{\partial x_l} = \sum_{k=1}^m \frac{\partial F(\vec{u})}{\partial u_k} \frac{\partial u_k}{\partial x_l}$. We need $\frac{\partial u_k(\vec{i}, j)}{\partial x_l}$ to compute $\nabla F$ and $\frac{\partial u_k(\vec{i}, j)}{\partial x_l}$ can be approximated by $\frac{u_k(\vec{i} + \mathbf{1}_l, j) - u_k(\vec{i}, j)}{x_{l,i_l+1} - x_{l,i_l}}$. Here, $\vec{i} + \mathbf{1}_l$ means to move the $l$-th coordinate of $\vec{i}$, namely $i_l$, to the next discretization point $i_{l+1}$. Repeating this type of calculations for the second-order derivatives $\nabla^2 H(\vec{u})$, the PDE in Equation 1 can be approximated in the following general form:

$$\vec{u}(\vec{i}, j+1) = \vec{u}(\vec{i}, j) + \delta_t Q(\{\vec{u}(\vec{i'}, j), \vec{i'} \in N(\vec{i})\}). \tag{4}$$

Here, we intentionally use function $Q$ to abstract out the actual form, because its derivation is pure arithmetic and is only marginally related to the main purpose of this paper. One important fact to notice: $Q$ depends on a *small* set of $\vec{u}(\vec{i'}, j)$'s, in which $\vec{i'}$ is a neighboring element of $\vec{i}$. We use $N(\vec{i})$ to represent the set of neighbors of $\vec{i}$. In the two applications we consider in this paper, $N(\vec{i})$ is actually small and only consists of the first and second order neighbors. Two elements are first or second order neighbors if and only if they differ in one dimension within two discretization points, or they differ in two dimensions, each dimension within one discretization point.

**Forward simulation.** By discretizing PDEs with finite difference approach (Equation 4), we can simulate future states of a physics system from an initial state by updating the left-hand side with values from the right-hand side of Equation 4 repeatedly. Interestingly, such a process can be

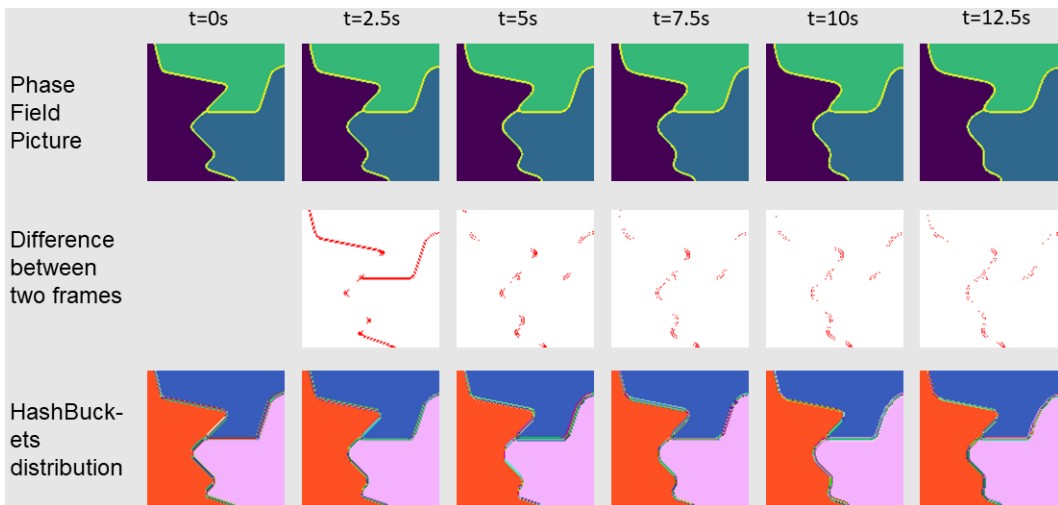

Figure 2: Intuitions that motivates LSH-SMILE. The updates of grain growth (difference between two consecutive frames, shown in the second row) only happens at the grain boundaries (show in red dots). Also, notice from the first and the second rows, elements with similar values of themselves and their neighbors share the same temporal dynamics. Hence in the third row, they can be hashed into the same hash bucket using LSH. The colors represent the hash buckets of the elements.

implemented as a multi-layer convolutional neural network (see, e.g., Xue et al. [2021]). See Figure 1 for a demonstration of the forward simulation process. Here we use the grain growth as an example.

**Backward Learning.** Backward learning allows us to learn the parameters of the PDE that governs the dynamics of a physics system from experiment data. It is developed with a series of work, e.g. Wen et al. [2021], Dupont et al. [2019], Xue et al. [2021] and Rubanova et al. [2019]. During backward learning, physics rules are expressed in the form of Equation 4 initialized with random parameters. The dataset consists of the observed pairs of states: $\vec{u}(\vec{p}, t)$ and $\vec{u}(\vec{p}, t + T)$, where $\vec{u}(\vec{p}, t)$ is the state of physics system at time $t$ and $\vec{u}(\vec{p}, t + T)$ is the state at time $t + T$. Here $T$ is a constant set manually. Starting with $\vec{u}(\vec{p}, t)$, a repeated evaluation of Equation 4 of $T$ times will yield the simulated state $\vec{u'}(\vec{p}, t + T)$. A loss function is defined to penalize the difference between the simulated state $\vec{u'}(\vec{p}, t + T)$ and observed state $\vec{u}(\vec{p}, t + T)$. In our experiment, we use the $L_2$ loss function. Back-propagation (Rumelhart et al. [1986], Robbins and Monro [1951]) is then applied to minimize the difference. Upon convergence, correct physics parameters are learned which yield the same temporal dynamics as the observed data.

**Locality Sensitive Hashing (LSH).** For a domain $S$ with distance measure $D$, a $LSH$ family is:

**Definition 1.** $\mathcal{H} = \{h : S \rightarrow U\}$ *is called a* $(r_1, r_2, p, q)$-*sensitive LSH function family for D if for any two points* $x, y \in S$, *one function* $h$ *chosen uniformly at random from* $\mathcal{H}$ *satisfies:*

- *if* $D(x, y) \leq r_1$, *then* $P_{h \in \mathcal{H}}[h(x)=h(y)] \geq p$,
- *if* $D(x, y) \geq r_2$, *then* $P_{h \in \mathcal{H}}[h(x)=h(y)] \leq q$.

In this paper, Euclidean distance is used as the distance measure $D$. In this case, a function in the $LSH$ hash function family has the following form $h_{\mathbf{a},b}(\mathbf{v}) = \left\lfloor \frac{\mathbf{a} \cdot \mathbf{v} + b}{r} \right\rfloor$, where $\mathbf{a}$ is a $d$-dimensional vector with entries chosen independently at random from a standard Gaussian distribution and $b$ is a real number chosen uniformly from the range $[0, r]$. $r$ is a hyper-parameter denoting the size of the hash bucket. We refer the reader to Datar et al. [2004] for the selection of $r$ and the resulting performance (i.e., the values of $r_1, r_2, p, q$).

## 3 Intuition

The intuitions behind the development of LSH-SMILE is inspired by the following three observations. Notice that these observations apply to a wide range of physics systems beyond the applications considered in this paper. Hence, we believe our method can be applied in a broad context.

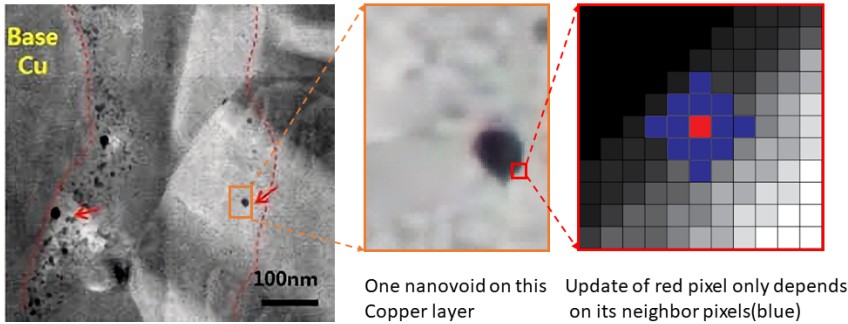

Figure 3: The spatial local dependency of the updates also motivates LSH-SMILE. The left picture (from Bremmert et al. [2019]) shows a nanovoid, enlarged in the middle. The update of one element of the nanovoid (shown in the red box) only depends on the values of itself and its neighboring elements (shown in the blue boxes in the right). This local dependency implies elements with similar values of themselves and their neighbors will share similar dynamics.

**Concentrated updates.** Our first observation is that most element updates are concentrated in small area. For example, consider our application of grain growth (Fan and Chen [1997]). As shown in Figure 2 (first and second rows), the updates are concentrated in rather small boundary area (shown in yellow in the first row). There are no updates inside the grains (shown in green, purple, blue of the first row). From our calculation, the boundary area where significant updates reside makes up under 1% of the entire image. Notice this phenomenon is prevalent in many physics systems, especially among the so-called *interface* problems. Such concentrated updates makes it wasteful to spread computation resources over the entire area. An algorithm focused on area of real updates can potentially save a lot of computation.

**Spatial local dependency.** Out second observation is that updating one element only depends on the values of the element itself and its small set of neighboring elements. Notice this property is a direct consequence of applying the finite difference method on PDEs. See, e.g., applying finite difference on Equation 1 results in Equation 4, in which the updates depend on a small set of elements in $N(\vec{i})$. As a consequence, this observation applies to a wide range of PDE systems. A more intuitive illustration is shown in Figure 3 for the nanovoid evolution application we consider. Here the update of one element (shown in the red box) depends on its first and second-order neighbors (shown in blue boxes).

**Elements share similar dynamics.** Our third observation is that many elements share similar temporal dynamics. Notice that this is a direct consequence of the spatial local dependency. Because the temporal update of one element depends on its own value and those of its neighbors, the temporal update of two elements will be the same if their own value and their neighbors' values meet. For example in Figure 2, the temporal dynamics of elements inside each grain are basically the same.

## 4  The LSH-SMILE Algorithm

Our proposed LSH-SMILE algorithm harnesses locality sensitive hashing to accelerate both the forward simulation and the backward learning of physics systems represented in PDEs. We first hash every element into a hash table based on LSH computed from the values of the element and its neighbors. Because of previous observations, the elements hashed into the same bucket share similar dynamics. The key idea behind LSH-SMILE is to handle *all* the elements in each hash bucket *at once*, hence reducing the complexity from proportional to the number of elements to proportional to the number of non-empty hash buckets, a drastic improvement. For example, the third row of Figure 2 colors elements of different hash buckets with different colors. We can see that elements inside each grain are hashed into a single hash bucket (shown in red, blue and pink), while elements on the boundary are hashed into many different hash buckets. In this way, only three operations suffice to handle the updates of *all* elements inside the three grains. In this section, we will first describe LSH-SMILE algorithm for forward simulation, then will discuss LSH-SMILE for backward learning. The pseudo code of LSH-SMILE for forward simulation is in Figure 4.

**Notation.** We first introduce the notation used in LSH-SMILE in Figure 4. $\mathcal{B}_t$ is the hash table at time $t$. $I$ is the initial state. $r, K, L$ are LSH parameters. $r_0$ is a threshold. $N$ is the number of simulation steps.

```
 1  Func Forward(N, I, r, K, L):                      1  Func MergeBuckets(B, K, L):
 2    B_0 ← EncodeInitState(I, K, L);                 2    B' = ∅;
 3    for t = 1, ..., N do                            3    for B ∈ B do
 4      B_t ← OneStep(B_{t-1}, r, K, L);              4      for l ∈ {1, ..., L} do
 5    end                                             5        if there exists B' ∈ B', such that
 6    return B_N;                                                B.LSH_l = B'.LSH_l then
 7  end                                               6          B' ← B' \ {B'};
 8  Func OneStep(B_t, r, K, L):                       7          B ← Merge(B, B');
 9    ActiveL ← ∅;                                    8        end
10    for B_{t,i} ∈ B_t do                           9      end
11      B_{t,i}.v_old ← B_{t,i}.v;                   10      B' ← B' ∪ {B};
12      B_{t,i}.v ←                                  11    end
13      Eval(B_{t,i}.v, NeighborV(i, B_t));          12    for B ∈ B' do
14      Update B_{t,i}.LSH_1, ..., .LSH_L;           13      for l ∈ {1, ..., L} do
15      if |B_{t,i}.v − B_{t,i}.v_old| > r_0 then    14        B.LSH_l ←
16        ActiveL ← ActiveL ∪ B_{t,i}.N                        LSH_l(B.v, NeighborV(B.rep, B'))
17      end                                          15      end
18    end                                            16    end
19    B_{t+1} ← B_t;                                 17    return B';
20    for a ∈ ActiveL do                             18  end
21      Remove a from the original bucket            19  Func EncodeInitState(I, K, L):
        that contains a in B_{t+1};                  20    B_0 = ∅;
22      B_a ←                                        21    for a ∈ I do
        CreateBucketElem(a, K, L);                   22      B_a ←
23      B_{t+1} ← B_{t+1} ∪ {B_a};                           CreateBucketElem(a, K, L);
24    end                                            23      B_0 ← B_0 ∪ {B_a};
25    B_{t+1} ← MergeBuckets(B_{t+1});               24    end
26    return B_{t+1};                                25    return MergeBuckets(B_0);
27  end                                              26  end
```

Figure 4: The forward simulation algorithm of $N$ steps using locality sensitive hashing. The main function is *Forward*, which calls children functions *OneStep*, *MergeBuckets*, and *EncodeInitState*.

**Data structure.** $\mathcal{B}_t$ is the hash table at time $t$, each entry of which is a hash bucket. In one hash bucket $B_{t,i} \in \mathcal{B}_t$, there is a representative value $B_{t,i}.v$, a representative element $B_{t,i}.rep$, a list of its elements' coordinates $B_{t,i}.P$ and a list of neighbor elements' coordinates $B_{t,i}.N$. Here we use union-find-delete data structure (Ben-amram and Yoffe [2011]) to organize $B_{t,i}.P$, which ensures that we can perform the union of the elements of two hash buckets and deleting one element from a hash bucket in nearly constant time. The idea of building such a hash bucket is that, using LSH, every element $\vec{u}(\vec{i}, j)$ that shares almost the same values for itself and for its neighbors $N(\vec{i})$ shall be hashed into the same bucket with high probability. We intentionally give all elements in one bucket a single value, namely $B_{t,i}.v$. We use $B_{t,i}.P$ to track all the elements in the bucket and $B_{t,i}.N$ to track all the elements that neighbor elements in $B_{t,i}.P$ but are not in $B_{t,i}.P$. According to Equation 4, the elements in one hash bucket have similar updates. Hence, we can update the value $B_{t,i}.v$ for all elements in this bucket following Equation 4 in one step, thus saving computation time.

**Encode initial state.** The first step of LSH-SMILE is to encode the input data into the data structure described above. At the beginning, since we do not have any prior information of the input data, we use a brute-force method which iterates all elements in the input data and construct a hash table where each hash bucket only contain one element. Then we perform *MergeBucket* which merges those buckets whose elements share colliding LSH codes.

**Forward and OneStep functions.** The main function for the forward simulation is *Forward*, which calls *OneStep* $N$ times to simulate forward simulation of $N$ time steps. Each *OneStep* simulates one step. In *OneStep*, first it iterates every hash bucket to perform an one-step update given in Equation 4. The update is carried out using function $Eval(\cdot)$. After the update, it updates its LSH hash code and sees if the update is bigger than a threshold $r_0$. If so, all elements in the bucket needs to be re-hashed. We do so by updating their LSH code. Notice this is an operation that is carried out *at once* for all elements in the bucket. Also notice that the value updates of the elements in the bucket may affect

the LSH codes of neighboring elements, namely those in $B_{t,i}.N$. We put these elements into *ActiveL* for later processing. After iterating all hash buckets, LSH-SMILE handles *ActiveL*, rehashing the elements inside one by one to make sure that they are in their correct buckets. Finally, it merges all the buckets with colliding LSH hash codes.

**Invariants.** LSH-SMILE strives to maintain the following invariants: (i) every element is in one and only one hash bucket. (ii) The LSH code computed for every element residing in a bucket collides with the LSH code of the representative element of the bucket. (iii) The LSH code of different hash buckets do not collide. (i) is guaranteed because we only move elements between buckets. (ii) is ensured because the change of the LSH code of one element can only happen if either the value of the element changes or the values of its neighboring elements change. When $r_0$ is set small, its own value change will trigger the exceeding of $r_0$ threshold and hence the element is rehashed. The values change of its neighbors will put the element in *ActiveL* and hence is rehashed. (iii) is guaranteed because of the *MergeBuckets* operation, since buckets with colliding LSH codes are merged.

**AND of OR LSH.** We introduce multi-probe LSH techniques from Lv et al. [2007] to guarantee high probabilities that two elements within the distance of $r$ will be hashed into the same hash bucket. Consider a series of LSH functions $h_{i,j}$ constructed in the form $h_{i,j}(\mathbf{v}) = \left\lfloor \frac{\mathbf{a}_{i,j} \cdot \mathbf{v} + b_{i,j}}{r} \right\rfloor$ where $a_{i,j}$ and $b_{i,j}$ are sampled in the way described in Section 2. Construct $g_i(p) = [h_{i,1}(p), \ldots, h_{i,K}(p)]$, select $L$ different functions $g_1, \ldots, g_L$. For one element $p$, hash $p$ into all $L$ buckets, denoted by $g_1, \ldots, g_L. p$ and $q$ "collides" if they collide under any of the $g_1, \ldots, g_L$ values. We refer to the next section on how to choose $K, L, r$ and the corresponding guarantees we can have.

**Merge Buckets.** During simulation, the values of the representative element and neighbors may change, leading to several hash buckets colliding on the LSH codes. To avoid duplicative computation, we use *MergeBuckets* to merge those buckets with colliding LSH codes.

**ActiveL.** The $ActiveL$ is a list of elements whose neighbors' values are updated which can potentially lead to their own LSH updates. We maintain this list during simulation and process these elements one by one in the end of *OneStep*.

**Backward learning.** The described forward simulation process can be embedded in the backward learning which learns the parameters of the PDEs from experimental data. The high-level idea is to harness stochastic gradient descend to adjust the PDE parameters so as to minimize the difference between the simulated results of $T$ steps and the experiment data after $T$ steps. Notice in the forward simulation, we do not retain the hash table for every time stamp. Otherwise there is a large overhead copying hash buckets between hash tables. However, elements values in previous time stamps are needed for back propagation. In this case, we slightly modify Equation 4, namely, replacing $\vec{u}(\vec{i'}, j)$ with $\vec{u}(\vec{i'}, j+1)$, when used in back-propagation.

# 5   Analysis

**Algorithm running time analysis.** Let $n$ be the size of input $I$, $b$ be the largest number of buckets in any $\mathcal{B}_t$, $g$ be the size of largest neighbor list for one bucket during simulation. The merging of two buckets $Merge(B, B')$ has to merge the element list $B.P$ and $B'.P$ as well as updating the neighbor's list. Merging element list is conducted using the union-find-delete data structure and is handled in near constant time. The bottleneck is to update the neighbor list, which scales $O(g \log g)$ using sorted list merge. For *MergeBuckets*, at most $bL$ *Merge* operations can happen and hence the complexity is $O(bKL + bLg \log g)$. In function $EncodeInitState(\cdot)$, line 21 to 24 iterates every element in $I$, line 25 calls $MergeBuckets$, thus this function have time complexity $O(n + bKL + bLg \log g)$. In function $OneStep(\cdot)$, line 10 to 18 iterates on $\mathcal{B}$, the time complexity is $O(bKL)$. Line 20 to 24 iterates on $ActiveL$, which has size smaller than $bg$. Line 25 calls $MergeBuckets$. Hence the time complexity of $OneStep$ is $O(bKL + bLg \log g)$. In function $Forward(\cdot)$, line 2 calls $EncodeInitState(\cdot)$, line 3 to 5 calls $OneStep(\cdot)$ for $N$ steps. The time complexity of $Forward(\cdot)$ is $O(bNKL + bNLg \log g + n)$. In comparison, a brute-force method has time complexity $O(nN)$. Our LSH-SMILE will be faster than the brute-force method when $bKL + bLg \log g << n$.

**Error bound introduced by LSH-SMILE**. When quantifying the approximation quality of LSH-SMILE, we assume all the elements during simulation are *naturally clustered*. Naturally clustered means that there exists $c > 1$ and $r > 0$, for any two elements $x$ and $y$, either they belong to the same

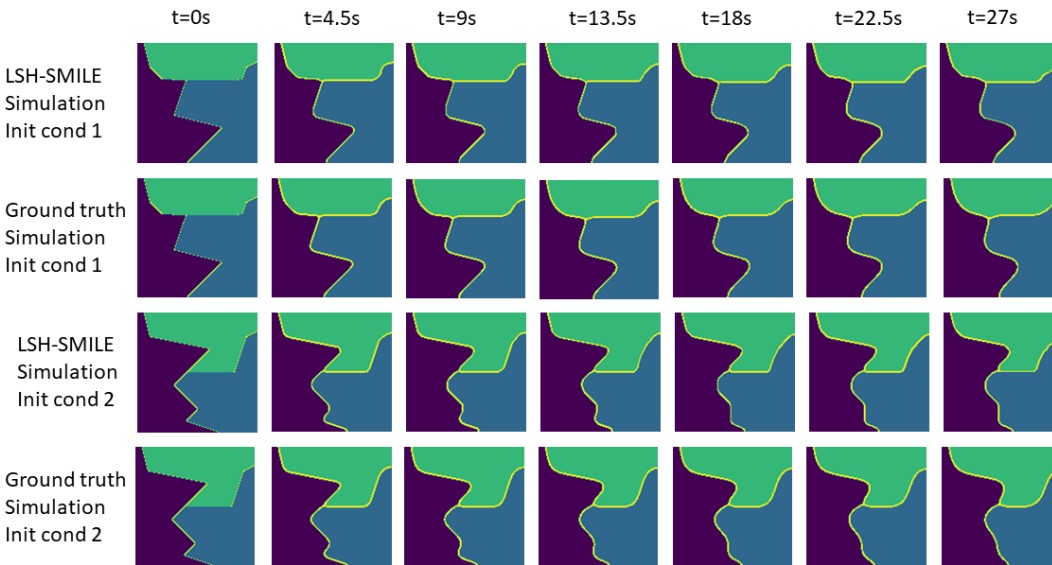

Figure 5: LSH-SMILE simulation results (the first and the third rows) closely match the ground truth (the second and the fourth rows). First two rows are initialized with condition 1. Last two rows are initialized with condition 2.

cluster and hence their distance $||x - y|| \leq r$, or they belong to different clusters and $||x - y|| \geq cr$. Here $x$ and $y$ refer to the vectors containing the values of the corresponding elements and their neighbors. Naturally clustering well describes the observed data in practice. In the grain growth example, the elements inside each grain share similar values (hence having small pairwise distances), while the elements of different grains or in the boundaries are different (hence having large pairwise distances). Under the natural clustering assumption, we can build a LSH function which guarantees with high probabilities that elements inside each cluster are hashed to the same bucket while elements of different clusters are hashed to different buckets, as reflected in the following theorem:

**Theorem 1.** *For $0 < \epsilon < 0.5$, let $C = \log_\epsilon(1 - \epsilon)$, then $0 < C < 1$. Suppose each $h_{i,j}$ is sampled from a $(r, cr, p, q)$-sensitive LSH function family. Pick $K = \max\{1, \lceil \log_{p/q} C \rceil\}$, $L = \frac{1}{\log_\epsilon(1 - p^K)}$. Build $L$ LSH functions in the way described in "AND of OR LSH", we have*

- *For all elements $x, y$ satisfying $||x - y|| \leq r$, i.e., they belong to the same cluster, we have $Pr(x \text{ collide with } y) \geq 1 - \epsilon$.*
- *For all elements $x, y$ satisfying $||x - y|| \geq cr$, i.e., they belong to different clusters, we have $Pr(x \text{ collide with } y) \leq \epsilon$.*

We will show the proof in the supplementary materials. A consequence of this theorem is to bound the approximation errors introduced in one-step LSH-SMILE simulation. Because of theorem 1, the pairwise distance among elements in one hash bucket is bounded by $r$ with high probability. LSH-SMILE uses the update of one representative element (shown in Equation 4) in replacement of the updates of all the elements in the bucket. Let us assume one step update based on Equation 4 magnifies this difference by $M$. In other words, for two elements whose distance bounded by $r$, the value distance of these two elements after one step update from Equation 4 becomes $Mr$. As a result, if we choose the parameters of the LSH functions according to theorem 1, we know after one call of $OneStep(\cdot)$, the errors will be bounded by $Mr$ with high probability (a union bound argument is needed). In experiment we found the magnitude of $M$ is around 10. We can hence control the error introduced by LSH-SMILE by setting $r$ to be small. We will show the forward simulation result in experiment to support this idea.

## 6 Experiments

**Forward simulation.** We first examine the proposed LSH-SMILE algorithm in forward simulation. We apply LSH-SMILE on two physics models for nano-structure evolution in materials. One is to model the grain growth (Fan and Chen [1997]), and the other is to model nanovoids evolution (Millett

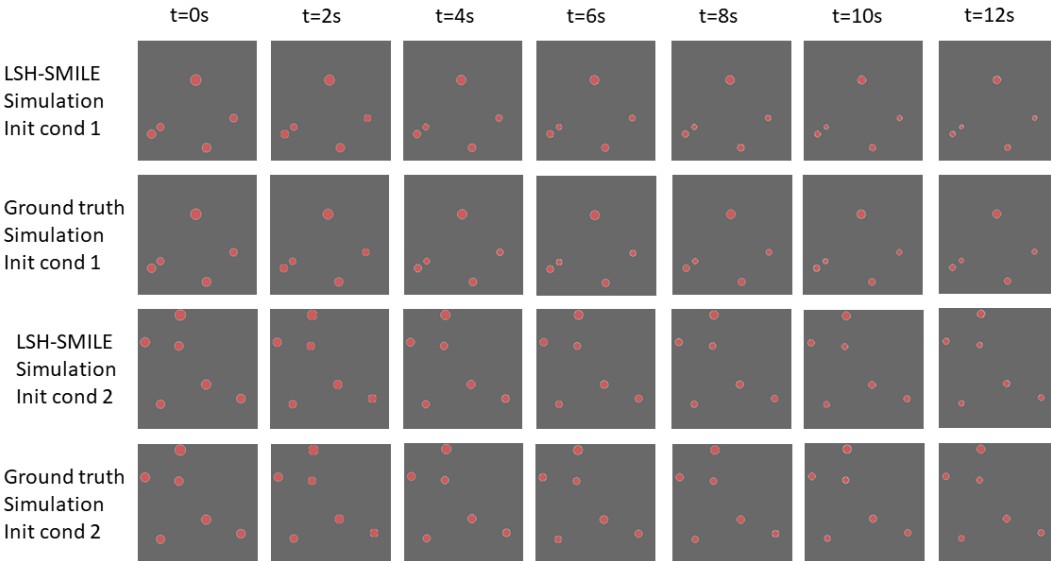

Figure 6: LSH-SMILE simulation results of nanovoids closely match the ground truth. The first two rows are initialized with condition 1. The last two rows are initialized with condition 2.

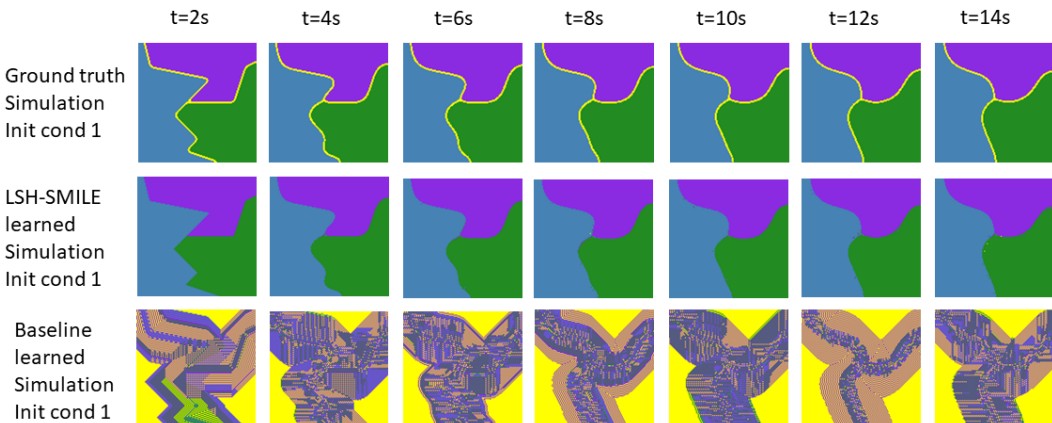

Figure 7: LSH-SMILE is able to learn the dynamics of grain growth, while the brute-force baseline cannot. The first row is the ground truth data. The second row shows the simulated dynamics of the physics model learned by LSH-SMILE, which matches the ground-truth well. The third row uses brute-force approach to learn, resulting in a physics model with incorrect dynamics.

et al. [2011]). In each experiment, two forward simulation algorithms are applied. One is the brute force method coded up using the Torch framework (Paszke et al. [2019] under Modified BSD license). The other is our LSH-SMILE simulation. The physics rules of the two applications are implemented in the $Eval(\cdot)$ function in the pseudo code in Figure 4.

**Grain growth simulation.** We evaluated the forward simulation for grain growth system (Fan and Chen [1997]). In this application, the $i$-th grain is represented using a phase field variable $\eta_i(\vec{p}, t)$. $\eta_i$ changes over time and evaluates to 1 when $\vec{p}$ inside the $i$-th grain and 0 outside the grain. The value of $\eta_i$ on the grain boundary is between 0 and 1. After discretizing the PDEs governing $\eta_i$'s dynamics using finite difference, the update rule for each grain is expressed in the following equation. More details can be found in the supplementary materials.

$$\eta_i(\vec{p}, t+1) = \eta_i(\vec{p}, t) - L_i dt \left( -A\eta_i(\vec{p}, t) + B\eta_i^3(\vec{p}, t) + 2\eta_i(\vec{p}, t) \sum_{i \neq j}^{N} \eta_j^2(\vec{p}, t) - \kappa_i \nabla^2 \eta_i(\vec{p}, t) \right).$$

| Algorithms | Runtime (hr) | Memory Usage(mb) |
|---|---|---|
| GG-Sim LSH-SMILE | **3.86** | **870** |
| GG-Sim Baseline | 16.84 | 1135 |
| NN-Sim LSH-SMILE | **2.13** | **2735** |
| NN-Sim Baseline | 53.13 | 3294 |
| GG-Sim Learn LSH-SMILE | **4.5** | **889** |
| GG-Sim Learn Baseline | 17.29 | 1212 |

Table 1: Running times of different approaches. LSH-SMILE needs less time and space compared to baseline methods.

Using the finite difference approach, $\nabla^2$ is expressed using a convolutional layer with the fixed 3x3 kernel [0 1 0, 1 -4 1, 0 1 0]. We set $dt$ to 0.05 in this simulation. The image size is 128 by 128. For LSH parameter, $r$ and $r_0$ are set to be 0.01, $K$ is 3 and $L$ is 10. The simulation results is in Figure 5, where LSH-SMILE simulates the physics process similarly compared to the ground truth computed by the brute-force approach. To measure the difference, we subtract the simulated result from LSH-SMILE with the ground truth. The difference is at the level of $10^{-5}$ after 180 steps.

**Nanovoid simulation.** We evaluated the forward simulation of nanovoid dynamics in materials under high temperature and irradiation (Millett et al. [2011]). The system is described using three phase field variables, $c_v(\vec{p}, t)$, $c_i(\vec{p}, t)$ and $\eta(\vec{p}, t)$. $c_v(\vec{p}, t)$ represents the fraction of void defects in unit volume of the material located at $\vec{p}$, while $c_i(\vec{p}, t)$ represents the fraction of interstitial concentration at $\vec{p}$. $\eta(\vec{p}, t)$ is the indicator function that evaluates to 1 if $\vec{p}$ is inside a void cluster, and 0 outside. The update function for field variable $c_v$ is:

$$c_v(\vec{p}, t+1) = c_v(\vec{p}, t) + dt M_v \nabla^2 \left( h(\eta) \frac{\partial f^s(c_v, c_i)}{\partial c_v} + j(\eta) \frac{\partial f^v(c_v, c_i)}{\partial c_v} - \kappa_v \nabla^2 c_v \right).$$

In this equation, $c_v$, $c_i$ and $\eta$ mean $c_v(\vec{p}, t)$, $c_i(\vec{p}, t)$ and $\eta(\vec{p}, t)$ when $(\vec{p}, t)$ are omitted. The actual definition of $f^s$, $f^v$ and the PDE equations for $c_i$ and $\eta$ are left to the supplementary materials. We start the simulation with two different starting state, shown in the first column of Figure 6. We set $dt$ to 0.1 in this simulation. The image size is 128 by 128. For LSH parameter, $r$ is set to be 0.0001, $K$ is 3 and $L$ is 10. The results is in Figure 6. It shows that LSH-SMILE simulate the physics process at comparable quality to the ground truth (difference at the level of $10^{-5}$ after 100 steps).

**Grain growth learning.** We examined the performance of LSH-SMILE in backward learning as well. The dataset is synthetic and contains 1700 frames of grain growth ground truth simulation results. The time and memory savings brought by LSH-SMILE allows the learning algorithm to match the predicted outcomes and the ground-truth outcomes that are $T$=30 steps away from the starting states, while the baseline method (embedding a brute-force forward approach) can only match the outcomes $T$=10 steps away, under the same computational budget. LSH-SMILE uses stochastic gradient descent, while the baseline use the Adam optimizer in our experiment. The ground truth parameter to be learned is all $L_i$'s=5.0, $A$=$B$=1.0, all $\kappa_i$'s=0.1. After 10 epochs of training, LSH-SMILE learned all $L_i$'s=11.6504, $A$=1.98483, $B$=2.01454, all $\kappa_i$'s=0.0834962, close to the ground-truth. At the same time, baseline method learned $L$=9.0462, $A$=1.9431, $B$=8.8226, $\kappa$=0.8431, far away from the ground-truth. We also simulated grain growth from the same initial condition using the parameters learned. The simulation is shown in Figure 7, verifying that LSH-SMILE learned parameters lead to similar dynamics as the groundtruth, while the baseline model learned an implausible model.

**Running time and memory comparison.** We examine the running time of both algorithms in simulation and learning. The forward simulation steps is $1700 \times 10 \times 30 = 510000$ for both baseline and LSH-SMILE. For the learning task, both algorithms train for 10 epoch. Our LSH-SMILE embeds forward simulation for $T = 30$ steps while baseline embeds $T = 10$ steps. The results is shown in Table 1. Here $GG$ stands for grain growth and $NN$ stands for nanovoid. We can see that our LSH-SMILE method drastically reduces the running time and memory usage in the same setting compared to baseline method implemented in Torch.

## 7 Conclusion

We propose LSH-SMILE, a unified framework to accelerate the forward simulation and backward learning of physics models, taking advantages of locality sensitive hashing. We show both theoretically and experimentally that LSH-SMILE simulates and learns physics models in a precise fashion and with reduced time and space complexity.

## Acknowledgements

This research was supported by NSF grants IIS-1850243, CCF-1918327. We thank anonymous reviewers for their comments and suggestions. C. S. acknowledges additional financial support for an internship at SenseTime. Y. X. discloses additional support Award No. W81XWH-18-1-0769 from the office of the assistant secretary of defense for health affairs.

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
