# OpenReview forum: "LSH-SMILE: Locality Sensitive Hashing Accelerated Simulation and Learning"
_NeurIPS.cc/2021/Conference — NeurIPS 2021 Poster_

### Official Review · Reviewer_z9La · 2021-07-15

**Rating:** 6
**Confidence:** 3

**Summary:**

The paper applies locality sensitive hashing to neural networks for solving Partial Differential Equations (PDEs). The paper proposes a solution called LSH-SMILE and claims several contributions: (1) LSH-SMILE achieves a drastic improvement in scalability than conventional approaches. (2) Theoretically, the paper proves a novel bound on the errors introduced by LSH-SMILE (3) Experimentally, the paper demonstrates LSH-SMILE exhibits comparable quality with exact approaches but costs less time and space.

**Limitations And Societal Impact:**

There is no sign of negative impact.

**Main Review:**

The paper is novel that applies Locality Sensitive Hashing (LSH) to physics systems for faster runtime. As a tradeoff, LSH introduces errors to the physic systems since LSH is an approximate approach. But the paper demonstrates that the errors are acceptable since the simulation quality of the physics system can match the quality of using conventional exact match approach, as shown in Figure 4 and Figure 5. Runtime-wise, the proposed method LSH-SMILE can outperform conventional approach by more than 10 times, as shown in Table 1.

Although the paper has merits in integrating computer science approaches with physics systems, it contains a lot of undefined physical terms that affect the understanding of the paper for general computer science audiences (like myself). The undefined physical terms impact reproducibility of the experiments as well. In particular, I suggest the authors improve the draft as follows:

(1)	Provide definitions for technical terms before using them, in plain language that general audiences can understand. For example, “forward simulation” and “backward learning” are introduced early in the Abstract, but they are not standard terms in computer science. Although computer science researchers often use “forward propagation” and “backward propagation” in neural networks, it is not sure whether the authors refer to the same things. There are a lot of undefined technical terms in the paper, such as “Partial Differential Equations”, “elements”, and “temporal dynamics” in the Abstract.

(2)	Provide toy examples for long definitions. The paper gives a definition for Partial Differential Equation (PDE) in Section 2 Preliminaries. The definition is as long as half a page, but the paper does not give any toy examples. It is unclear what a PDE looks like.

(3)	Specify the inputs and outputs of the LSH algorithm. The input format of an LSH algorithm should be a query vector and a set of item vectors. The output format should be the most similar item vectors to the query vector. When applying LSH to physics systems, it is natural that one has to formulate physics systems into query vector and item vectors. However, the paper does not illustrate how the formulation looks like. It will be good if the authors can add the details to the paper for better clarity.


**Time Spent Reviewing:**

4

---

> ### Author Response · Authors · 2021-08-09
> **Response to Reviewer z9La**
>
> Thank you for the reviews. We make clarifications on the following points.
>
> To all the reviewers:
>
> -- 1. Applicability of LSH-SMILE on other problems.
>
> LSH-SMILE is not limited to the two physics problems considered in the paper. The applicability of locality-based hashing depends on three key traits. The first is that only a small fraction of elements change while the majority of elements remain the same in one step forward simulation. The second is that the one-step update of one element depends on the values of a small set of neighboring elements. The third is that many elements share similar temporal dynamics. The three traits are shared among the simulations of a wide variety of physics phenomena, e.g., in fluid dynamics, heat transfer, cracking formation, etc. Due to this reason, we anticipate that LSH-SMILE can accelerate the forward simulation and backward learning in these domains as well. Indeed, our paper has been focused on microstructure evolution in materials, but we will expand to other applicational domains in future work.
>
> -- 2. Innovation beyond applying LSH to physics simulation
>
> Despite LSH has been developed in previous research, the application of LSH in physics simulation and learning is different from its classical application in nearest neighbor search and has resulted in novel challenges. One crucial difference is that the hash table remains the same in the nearest neighbor search and previous analysis on LSH has been focused on the collision probability of one query element and its nearest neighbor in the static hash table. However, in our application, we need to maintain a hash table that changes dynamically over time. This leads to the new LSH-SMILE algorithm which drastically differs from previous LSH-based algorithms.
>
> Reviewer#4:
>
> Improvement:
>
> 1.	Thank you for providing this suggestion. We will define related terms more carefully in the next version of the paper. We plan to give definitions on all technical terms in section 2.
>
> 2.	Thank you for providing this suggestion. We plan to use the simplified grain growth model as a toy example to explain what the PDE equations look like, as well as explaining the loss function in backward computation, etc.
>
> 3.	Thank you for providing this suggestion. We plan to give an example (using the grain growth model) in the description of the LSH-SMILE algorithm to specify the size of input, output. See our response (point 2) to all the reviewers. Our usage of LSH is different from its standard usage that returns the nearest neighbor of a query vector. Because of this, the input to the LSH algorithm is not a query vector, as in the conventional case. Instead, we need to maintain that all vectors reside in their correct hash buckets dynamically over time. Hence, the input of our LSH-SMILE algorithm is the entire hash table before a one-step update; and the output is the entire hash table again after a one-step update.

---

> > ### Comment · Reviewer_z9La · 2021-08-18
> > **Thank you for the author response**
> >
> > Thank you for the author response. I would like to see the paper accepted since the application of LSH algorithm to physic system is novel. But if it is not accepted, I encourage the authors to continue the research and submit the paper to other venues.

---

### Official Review · Reviewer_z4Ec · 2021-07-16

**Rating:** 6
**Confidence:** 4

**Summary:**

This paper proposes a novel LSH based approach for efficient forward and backward pass of Partial Differential Equations (PDEs). The proposed LSH-SMILE approach focus on local updates and use LSH to find the local neighbors. Moreover, LSH-SMILE propose a bucket merging method to avoid empty buckets.

**Limitations And Societal Impact:**

Suggestions:

LSH is a great approach for scaling up large scale machine learning models. Is is possible to demonstrate more experiments on larger scale?

**Main Review:**

Strength:
1. Novel formulation that transforms PDEs into the nearest neighbor problem
2. Efficient combination of LSH and PDEs.
3. Error bounds for the errors

**Time Spent Reviewing:**

12

---

> ### Author Response · Authors · 2021-08-09
> **Response to Reviewer z4Ec**
>
> Thank you for the reviews. We make clarifications on the following points.
>
> To all the reviewers:
>
> -- 1. Applicability of LSH-SMILE on other problems.
>
> LSH-SMILE is not limited to the two physics problems considered in the paper. The applicability of locality-based hashing depends on three key traits. The first is that only a small fraction of elements change while the majority of elements remain the same in one step forward simulation. The second is that the one-step update of one element depends on the values of a small set of neighboring elements. The third is that many elements share similar temporal dynamics. The three traits are shared among the simulations of a wide variety of physics phenomena, e.g., in fluid dynamics, heat transfer, cracking formation, etc. Due to this reason, we anticipate that LSH-SMILE can accelerate the forward simulation and backward learning in these domains as well. Indeed, our paper has been focused on microstructure evolution in materials, but we will expand to other applicational domains in future work.
>
> -- 2. Innovation beyond applying LSH to physics simulation
>
> Despite LSH has been developed in previous research, the application of LSH in physics simulation and learning is different from its classical application in nearest neighbor search and has resulted in novel challenges. One crucial difference is that the hash table remains the same in the nearest neighbor search and previous analysis on LSH has been focused on the collision probability of one query element and its nearest neighbor in the static hash table. However, in our application, we need to maintain a hash table that changes dynamically over time. This leads to the new LSH-SMILE algorithm which drastically differs from previous LSH-based algorithms.
>
> Reviewer#3:
>
> Limitation:
>
> 1.	Please refer to our response (point 1) to all reviewers. Thank you for the suggestion. We will run larger-scale experiments on a wide variety of physics systems.

---

### Official Review · Reviewer_nWzM · 2021-07-20

**Rating:** 6
**Confidence:** 2

**Summary:**

In learned Partial Differential Equation (PDE) solvers, the forward simulation (thereby data collection) and backward learning processes are very time consuming. This paper tries to mitigate this issue by using Locality Sensitive Hashing (LSH).

The motivation for this approach comes from three critical observations -
1) that most PDE updates in a close vicinity tend to be similar ,i.e., in a one step update, most elements change only on few steps and remain unchanged in the rest of the process.
2) one step updates for a element is dependent on the neighborhood (neighborhood can be thought of a context in NLP).
3) most points share similar temporal dynamics

In learned PDE, the practical simulation happens by discretizing finite-difference approach and obtaining learnable vectors.

Starting from an initial state and hashing to one bucket per element, LSH-Smile iteratively pools elements that have similar hash codes into the same buckets. On each hash bucket, there will be a classic PDE update (same update for all elements in a bucket). Then the hash codes of buckets are updated to  obtain the ones with significant change (>r, a threshold). Then the elements in those buckets are rehashed.

The backward process is similar with an additional gradient computation step that I presume is also done a bucket basis.

I’m not entirely sure abt the input structure and the hash function used (I may have missed some details).

But essentially, LSH-SMILE does the standard updates process on a chunk by chunk basis instead of on all elements.






**Ethical Concerns:**

No evident ethical concerns

**Limitations And Societal Impact:**

Limitations:

1. LSH theory is a direct import from prior literature (understandable though). The primary novelty is integrating it with the PDE solver's forward and backward passes.

2. There maybe PDE applications where the three critical observations might not hold.

3. Noticeable typos like

line 183 : perform update ---> perform an update
line 189 : every elements --> every element
line 257 : by we directly coding --> because we directly coded
line 258 : rules is --> rules are

**Main Review:**

Strengths

1. Neat idea
2. Impressive gains in time and memory with negligible difference with ground truth simulations.

**Time Spent Reviewing:**

2

---

> ### Author Response · Authors · 2021-08-09
> **Response to Reviewer nWzM**
>
> Thank you for the reviews. We make clarifications on the following points.
>
> To all the reviewers:
>
> -- 1. Applicability of LSH-SMILE on other problems.
>
> LSH-SMILE is not limited to the two physics problems considered in the paper. The applicability of locality-based hashing depends on three key traits. The first is that only a small fraction of elements change while the majority of elements remain the same in one step forward simulation. The second is that the one-step update of one element depends on the values of a small set of neighboring elements. The third is that many elements share similar temporal dynamics. The three traits are shared among the simulations of a wide variety of physics phenomena, e.g., in fluid dynamics, heat transfer, cracking formation, etc. Due to this reason, we anticipate that LSH-SMILE can accelerate the forward simulation and backward learning in these domains as well. Indeed, our paper has been focused on microstructure evolution in materials, but we will expand to other applicational domains in future work.
>
> -- 2. Innovation beyond applying LSH to physics simulation
>
> Despite LSH has been developed in previous research, the application of LSH in physics simulation and learning is different from its classical application in nearest neighbor search and has resulted in novel challenges. One crucial difference is that the hash table remains the same in the nearest neighbor search and previous analysis on LSH has been focused on the collision probability of one query element and its nearest neighbor in the static hash table. However, in our application, we need to maintain a hash table that changes dynamically over time. This leads to the new LSH-SMILE algorithm which drastically differs from previous LSH-based algorithms.
>
> Reviewer#2:
>
> Limitations:
>
> 1.	Please see our response 2 to all reviewers.
>
> 2.	Thank you for raising this point. Please see our response 1 to all reviewers. We admit that there exist cases that do not satisfy the three traits. However, the simulations of a large set of physics phenomena satisfy these three traits.
>
> 3.	Thank you for pointing out our typos in the paper. We will correct all typos in the next version of the paper.

---

### Official Review · Reviewer_gTTS · 2021-07-23

**Rating:** 6
**Confidence:** 3

**Summary:**

The paper proposes a method (LSH-SMILE) that applies LSH to speed up PDE forward simulation and backward learning.
Observing that most of the elements are very similar, they can be hashed into buckets and each bucket can be represented by a single value, thus reducing computational and memory cost.
The method is validated empirically on two problems, grain growth and nanovoid simulation, showing that it is much faster than standard brute-force computation while maintaining reasonable accuracy.

**Limitations And Societal Impact:**

Yes

**Main Review:**

Strengths:
- The paper is well-written and easy to understand. Section 3 provides valuable intuition, and I enjoyed reading it.
- The method works for both forward simulation and backward learning, by leveraging the nature of differential equations (instead of relying on auto-differentiation, which can be memory-costly).
- The method can significantly speed up the PDE simulation and learning.

Weaknesses:
1. Limitation of application: it's not clear to me how widely the proposed method can be apply, as there's no discussion of its limitations. It seems to work for two narrow physics problem. However, I'm not sure which conditions will allow it to work well. This is the most significant weakness.
2. It seems that the algorithm is hard to parallelize, compared to the brute-force simulation. Thus the brute-force computation may able to better use GPUs to speed it, while LSH-SMILE would be bound to CPUs?

Questions:
1. The observation that close-by elements are very similar, is quite reminiscent of fast multi-pole method. I wonder if there's a connection here.
2. In Figure 3, Func OneStep, line 15, the ActiveL set is updated when the change is larger than r. However, what if the change per iteration is smaller than r but over time it can add up to be larger than r. Would the ActiveL set be updated then?

============== Post-rebuttal

Thank you for the response to my questions. That has helped me understand the method better.

The clarification on the setting where the method is expected to work is valuable. Given that the goal of the paper is to accelerated simulation and learning on physical systems in general, I think the case could be made much stronger if the applications are more diverse.

I've increased my rating.

**Time Spent Reviewing:**

3

---

> ### Author Response · Authors · 2021-08-09
> **Response to Reviewer gTTS**
>
> Thank you for the reviews. We make clarifications on the following points.
>
> To all the reviewers:
>
> -- 1. Applicability of LSH-SMILE on other problems.
>
> LSH-SMILE is not limited to the two physics problems considered in the paper. The applicability of locality-based hashing depends on three key traits. The first is that only a small fraction of elements change while the majority of elements remain the same in one step forward simulation. The second is that the one-step update of one element depends on the values of a small set of neighboring elements. The third is that many elements share similar temporal dynamics. The three traits are shared among the simulations of a wide variety of physics phenomena, e.g., in fluid dynamics, heat transfer, cracking formation, etc. Due to this reason, we anticipate that LSH-SMILE can accelerate the forward simulation and backward learning in these domains as well. Indeed, our paper has been focused on microstructure evolution in materials, but we will expand to other applicational domains in future work.
>
> -- 2. Innovation beyond applying LSH to physics simulation
>
> Despite LSH has been developed in previous research, the application of LSH in physics simulation and learning is different from its classical application in nearest neighbor search and has resulted in novel challenges. One crucial difference is that the hash table remains the same in the nearest neighbor search and previous analysis on LSH has been focused on the collision probability of one query element and its nearest neighbor in the static hash table. However, in our application, we need to maintain a hash table that changes dynamically over time. This leads to the new LSH-SMILE algorithm which drastically differs from previous LSH-based algorithms.
>
> Reviewer#1:
>
> Weakness:
>
> 1. See above for the applicability of LSH-SMILE on other problems.
>
>
> 2. Parallelization of LSH-SMILE.
>
>     Thank you for this comment. In fact, LSH-SMILE can be parallelized because the updates in different regions are mutually independent. For example, the updates for the pixels at the upper-left corner are independent of those at the lower-right corner if they reside in different hash buckets. Because one-step updates may trigger changes for neighboring pixels, the algorithm requires an additional locking mechanism to ensure the validity of neighboring pixels. However, the updates will not affect pixels that are many pixels apart. In a nutshell, parallelization can be implemented for LSH-SMILE given a careful design of the locking mechanism.
>
>     We briefly describe our preliminary idea. First in the OneStep function, since the computation of lines 10 to 17 is independent for each bucket, we can use different threads to complete the computations for different buckets. Second in lines 20 to 25 the OneStep function, we will implement a locking mechanism to handle computation which involves multiple buckets (lines 21 to 23). The locking mechanism works like this: the thread first tries to obtain the lock of the bucket before accessing it. It unlocks the bucket after the involved computation finishes. If a thread tries to lock a bucket but finds out that it’s locked, the thread will target another unlocked bucket.
>
> Questions:
>
> 1.  Thank you for raising this question. We will discuss the connections between our approach and the fast multi-pole method (FMM) in the next version of the paper. We read FMM and find that it is used to solve long-ranged forces in the n-body problem. Our understanding is that FMM groups elements to be considered based on their distance to each other. However, LSH-SMILE groups up elements based on their similarity in future updates. We believe this is a key difference. Meanwhile, FMM and LSH-SMILE are solving different problems. FMM focuses on calculating forces between elements while LSH-SMILE aims to simulate the temporal dynamics of elements.
>
> 2.  Thank you for raising this question. The general idea is that OneStep function runs one time per iteration. Inside the OneStep function, the ActiveL set is set to be empty at the beginning (line 9). It is adjusted dynamically if there is a bucket whose value change is larger than r. These changes won’t accumulate over iterations.

---

### Decision · Program_Chairs · 2021-09-27

**Decision:**

Accept (Poster)

**Comment:**

All the reviewers concurred that this paper is above bar for publication. Rebuttal reaffirmed that sentiment. Reviewers like the idea of using LSH for speeding up simulations.  In particular,  applying LSH algorithm to a completely new domain, i.e. physic systems. The results look promising and since the paper belongs to interdisciplinary research, reviewers agree that even though the paper is less advanced in algorithmic techniques, it will be a educative read for a broader community of people.

Please take in account comments from the reviewers to improve the paper.